# The Dynamics of Respiratory Microbiota during Mechanical Ventilation in Patients with Pneumonia

**DOI:** 10.3390/jcm9030638

**Published:** 2020-02-27

**Authors:** Seongji Woo, So-Yeong Park, Youngmi Kim, Jin Pyeong Jeon, Jae Jun Lee, Ji Young Hong

**Affiliations:** 1Institute of New frontier Research, Hallym University College of Medicine, Chuncheon 24253, Korea; seong-jikr@nate.com (S.W.); qkr94006@hanmail.net (S.-Y.P.); kym8389@hanmail.net (Y.K.); jjp6553@hallym.or.kr (J.P.J.); iloveu59@hallym.or.kr (J.J.L.); 2Department of Neurosurgery, Chuncheon Sacred Heart Hospital, Hallym University Medical Center, Chuncheon 24253, Korea; 3Division of Pulmonary and Critical Care Medicine, Department of Medicine, Chuncheon Sacred Heart Hospital, Hallym University Medical Center, Chuncheon 24253, Korea; 4Lung Research Institute of Hallym University College of Medicine, Chuncheon 24253, Korea

**Keywords:** mechanical ventilation, pneumonia, microbiome

## Abstract

Bacterial pneumonia is a major cause of mechanical ventilation in intensive care units. We hypothesized that the presence of particular microbiota in endotracheal tube aspirates during the course of intubation was associated with clinical outcomes such as extubation failure or 28-day mortality. Sixty mechanically ventilated ICU (intensive care unit) patients (41 patients with pneumonia and 19 patients without pneumonia) were included, and tracheal aspirates were obtained on days 1, 3, and 7. Gene sequencing of 16S rRNA was used to measure the composition of the respiratory microbiome. A total of 216 endotracheal aspirates were obtained from 60 patients. A total of 22 patients were successfully extubatedwithin3 weeks, and 12 patients died within 28days. Microbiota profiles differed significantly between the pneumonia group and the non-pneumonia group (Adonis, *p* < 0.01). While α diversity (Shannon index) significantly decreased between day 1 and day 7 in the successful extubation group, it did not decrease in the failed extubation group among intubated patients with pneumonia. There was a significant difference in the change of βdiversity between the successful extubation group and the failed extubation group for Bray-Curtis distances (*p* < 0.001). At the genus level, *Rothia*, *Streptococcus*, and *Prevotella* correlated with the change of β diversity. A low relative abundance of *Streptococci* at the time of intubation was strongly associated with 28-day mortality. The dynamics of respiratory microbiome were associated with clinical outcomes such as extubation failure and mortality. Further large prospective studies are needed to test the predictive value of endotracheal aspirates in intubated patients.

## 1. Introduction

Pneumonia is a leading cause of hospitalization and death and is associated with considerable medical costs among adults in the United States [1,2]. Intensive care unit admission of older patients with pneumonia is associated with improved survival, without a significant increase in costs [3]. The most recent guidelines recommend that the antibiotic regimen should be adjusted on the basis of respiratory cultures [4]. However, culture-based analysis has low sensitivity and a long turnaround time [2]. Furthermore, in 30–78% of patients with pneumonia, no bacterial pathogen is isolated [2,5,6]. Next-generation sequencing of microbial DNA in patient samples is expected to overcome the limitations of culture-based diagnosis. Sequencing of the 16S rRNA gene enables identification of anaerobes and fastidious organisms.

Mechanical ventilation is associated with dysbiosis of the respiratory microbiome, and respiratory microbiota markers may predict the development of ventilator-associated pneumonia or mortality in ICU patients [7,8,9]. Investigating the impact of respiratory microbiota on clinical outcomes or its association with the host response may lead to the development of novel microbiome-targeted therapeutics to replace the current culture-based practice [10]. Moreover, next-generation sequencing technologies for the diagnosis of pneumonia tools in ICU patients are under investigation.

We hypothesized that the composition of the respiratory tract bacterial microbiome during mechanical ventilation would be predictive of the clinical outcomes (extubation failure and 28-day mortality) in intubated patients with pneumonia. We evaluated changes in the microbiota in endotracheal aspirates (ETA) from mechanically ventilated patients by performing 16S rRNA sequencing.

## 2. Materials and Methods

### 2.1. Subjects and Study Design

This prospective study was conducted in the ICU of Chuncheon Sacred Heart Hospital, South Korea. The participants consisted of 41 and 19 patients with and without pneumonia, respectively. The patients were placed on mechanical ventilation at the time of ICU admission from July 2017 to August 2018 and were prospectively followed up. The exclusion criteria were: (1) age < 18 years; (2) initiation of mechanical ventilation ≥ 48 h after ICU admission; (3) presence of neuromuscular disease, such as amyotrophic lateral sclerosis; (4) inability to provide informed consent; and (5) duration of mechanical ventilation < 7 days. The diagnosis of pneumonia was based on the international guidelines. Clinical data including demographic variables, comorbidities, indication for intubation, and severity of illness (Acute Physiology and Chronic Health Evaluation (APACHE II) score and Sequential Organ Failure Assessment (SOFA) score) were recorded. The clinical outcomes included the success of weaning from mechanical ventilation at 3 weeks and the 28-day mortality rate.

Informed consent was obtained from patients or their surrogates. The study protocol was approved by the Institutional Review Board of Chuncheon Sacred Heart Hospital (IRB approval number: 2017-47).

### 2.2. DNA Extraction PCR and Sequencing

Sampling was performed by endotracheal aspiration. Initial samples were collected within 24 h of initiation of mechanical ventilation; sequential sample collection was performed on days 3 and 7 thereafter. Total DNA was extracted from 200 μL of 235 ETA samples using a commercial microbial DNA isolation kit (Qiagen, Hilden, Germany).

The extracted DNA was amplified using primers targeting the V3 to V4 regions of the prokaryotic 16S rRNA gene.

The primers were:

16S_V3_F(5′-TCGTCGGCAGCGTCAGATGTGTATAAGAGACAGCCTACGGGNGGCWGCAG-3′)

16S_V4_R(5′GTCTCGTGGGCTCGGAGATGTGTATAAGAGACAGGACTACHVGGGTATCTAATCC-3′)

The amplification program was as follows: 3 min at 95 °C (denaturation); 21 cycles of 30 s at 94 °C (denaturation); 30 s at 58 °C (annealing); 30 s at 72 °C (elongation); and 5 min at 72 °C (final extension). Amplicons were quantified and sequenced on a MiSeq (Illumina, San Diego, CA, USA), in accordance with the manufacturer’s instructions.

### 2.3. Data Processing

FASTQ files of the MiSeq raw data were generated by classifying different samples using the index sequences. Adaptor sequences were removed, and error correction was performed using SeqPurge. Paired-end reads obtained from the MiSeq system were merged using FLASH software. Merged sequences of <400 bp were removed. Sequencing errors (i.e., low-quality, ambiguous, and chimeric sequences) were trimmed using CD-HIT-OUT. Operational taxonomic units (OTUs) were formed by clustering over 97% of similar sequences. Taxonomic assignment was carried out using BLASTN (v. 2.4.0), with the National Center for Biotechnology Information 16S microbial database as reference, using information for the most similar species. Taxonomy was not defined when the query coverage of the best hit that matched the database was <85%, and the identity of the match area was <85%. An average of 122,818 reads per sample was obtained (minimum, 4; maximum, 587,337), and 19 samples with a small number of reads (<2500) were filtered out for quality control. Finally, 216 samples were analyzed.

### 2.4. Statistical Analysis

Biodiversity and community similarity analyses were performed using MD healthcare (Seoul, South Korea). Microbial community comparisons were performed using Quantitative Insights in Microbial Ecology software (QIIME, v. 1.8) and the R package Vegan [11]. To test for significant associations, the Mann–Whitney U test was used for continuous variables, and the chi-squared test was used for categorical variables. The α-diversity of the microbiota was measured by calculating the Shannon diversity index and the Simpson index. The β-diversity index was defined as the extent of similarity between microbial communities based on the degree of shared structure. The principal coordinates analysis (PCoA) was based on the Bray–Curtis similarity matrix of the square-root-transformed relative abundance of bacterial genera. Adonis tests were performed to confirm the differences between groups, and logistic regression was used to assess differences in the change in principal coordinates between the two groups.

Linear discriminant analysis effect size (LEfSe) was used to detect unique biomarkers by determining the relative abundance of bacterial taxa. A linear discriminant analysis (LDA) score of >2.5 was used to identify features that significantly discriminated among groups [12]. The Igraph [13] and centiserve [14] packages for R were used to analyze interactions within the pulmonary microbial communities of intubated patients. Predictive functional analysis was performed using Tax4Fun software [15]. Functional capabilities of the microbial communities were predicted by mapping 16S rRNA gene sequences and performing functional annotation based on the Kyoto Encyclopedia of Genes and Genomes. Receiver operating characteristic (ROC) curves were generated to assess the performances of microbial biomarkers. Statistical analysis was performed with R version 3.4.4 for Windows™ 10; *p*-values < 0.05 were considered to indicate statistical significance.

## 3. Results

### 3.1. Characteristics of the Study Population

This study analyzed samples from 41 patients with pneumonia and 19 patients without pneumonia. In total, 216 samples were analyzed by 16S rRNA sequencing, which yielded 30,236,835 sequences (median, 123,541; interquartile range, 97,015–156,914 sequences per sample). Of the 216 samples, 2883 OTUs were detected and classified into 37 phyla and 1085 genera. The baseline characteristics of the study population are listed in Table 1.

The pneumonia and non-pneumonia groups did not differ in terms of age, sex, Charlson comorbidity index, or severity indexes (APACHE and SOFA scores). The proportion of patients with acute respiratory distress syndrome and the causes of intubation were significantly different between the two groups. The patients who were placed on mechanical ventilation due to pneumonia were divided into two groups according to the clinical outcome: successful or failed extubation within 3 weeks. The successful and failed extubation groups were similar in terms of age, sex, and Charlson comorbidity index. The severity index and mortality rate were higher in the failed extubation group than in the successful extubation group.

### 3.2. Day 1 ETA Microbiome Composition in Intubated Patients

The predominant bacterial phyla were *Proteobacteria* and *Firmicutes* (Figure 1a). Overall, 77% of reads belonged to one of these phyla (pneumonia, 78%; non-pneumonia, 76%). The predominant genera were defined as those that comprised > 0.5% of the total DNA sequences; 20 predominant genera were detected.

The species richness was similar in the pneumonia and non-pneumonia groups, as shown by the Shannon index (*p* = 0.24) and Simpson index (*p* = 0.37). The β-diversity determined by PCoA differed significantly between the two groups (Figure 1b, Adonis *p* = 0.001). LEfSe analysis revealed 14 discriminating taxa. At the genus level, *Pseudomonas*, *Corynebacterium*, *Veillonella*, *Rothia*, *Enterococcus*, and *Neisseria* predominated in the pneumonia group; *Streptococcus*, *Prevotella*, *Alloprevotella*, *Granulicatella*, and *Mycoplasma* predominated in the non-pneumonia group.

Among patients with pneumonia, the α-diversity did not differ between the successful extubation and failed extubation groups (Figure 2a). The β-diversity differed between the two groups at day 1 (Figure 2b, r^2^ = 0.0408, *p* = 0.021). Among pneumonia patients, *Bacteroidetes* was significantly more abundant in the successful extubation group than in the failed extubation group (8.5% vs. 6.7%). At the genus level, LEfSe analysisrevealed significantly higher relative abundances of *Haemophilus* and *Streptococcus* and a lower abundance of *Enterococcus* in the successful extubation group, compared with the failed extubation group (Figure 2c).

### 3.3. Changes in Respiratory Microbiota during Mechanical Ventilation in Patients with Pneumonia

Changes in diversity over time differed between the successful extubation and failed extubation groups (Figure 3). The Shannon index significantly decreased between days 1 and 7 in the successful extubation group, but not in the failed extubation group. In addition, intra-individual Bray–Curtis dissimilarities between days 1 and 7 were significantly different in the successful extubation group, but not in the failed extubation group.

There was a significant difference in the change in PCo2 between the successful extubation and failed extubation groups (Adonis, *p* = 0.001). The abundances of *Rothia*, *Streptococcus*, and *Prevotella* were positively correlated with the change in PCo2.

Ecological interactions within pulmonary communities of intubated patients with pneumonia were evaluated (Figure 4). The network had a diameter of 52, with an average clustering coefficient of 0.11 and an average number of neighbors of 2.85. The microbial network of intubated patients with pneumonia consisted of 53 nodes and 60 edges. *Streptococcus* demonstrated negative associations with *Acinetobacter* and *Corynebacterium*. *Streptococcus* had the highest connectivity, indicating a cooperative strategy with *Rothia*, *Gemella*, *Granulicatella*, *Prevotella*, and *Actinomyces*.

Predictive functional profiling showed that the successful extubation group had a greater proportion of bacterial genes related to mismatch repair and homologous recombination, compared to the failed extubation group. In contrast, the failed extubation group had greater proportions of genes related to nitrogen metabolism and porphyrin and chlorophyll metabolism (Appendix A).

### 3.4. Influence of 28-Day Mortality on Respiratory Microbiota

The α-diversity did not differ between surviving and dead patients after 28 days, but β-diversity differed at day 1 (r^2^ = 0.0436, *p* = 0.008). The abundances of *Streptococcus*, *Haemophilus*, and *Neisseria* were enriched in the 28-day survival group (LDA score > 2.5, Figure 5). The 28-day death group exhibited higher relative abundances of *Corynebacterium* and *Alloprevotella*, compared to the 28-day survival group.

The predicted probability of 28-day mortality based on *Streptococcus* abundance had an area under the ROC curve of 0.74, with 86% sensitivity and 63% specificity. This was greater than the area under the ROC curve of the APACHE score (0.628). At day 1, *Streptococcus* had a relative abundance of < 9.6% in 21/38 (55.3%) of patients in the 28-day survival group and 12/14 (85.7%) of patients in the 28-day death group.

## 4. Discussions

We evaluated longitudinal changes in the respiratory microbiome of intubated patients who were placed on mechanical ventilation at the time of admission to the ICU. Consistent with previous reports [6,9,16], a few taxa dominated the respiratory microbiome of intubated patients; 16S rRNA sequencing enabled identification of atypical organisms in patients with negative culture results. Furthermore, mechanically ventilated patients with pneumonia tended to have lower α-diversity than those without pneumonia; bacterial α-diversity also tended to decrease during mechanical ventilation, as reported previously, but this difference was not statistically significant [8,16].

The β-diversity (Bray–Curtis dissimilarity) significantly differed between the pneumonia and non-pneumonia groups (Figure 1). We found increased abundances of *Pseudomonas*, *Corynebacterium*, and *Rothia*, as well as decreased abundances of *Streptococcus* and *Prevotella* in intubated patients with pneumonia, compared to those without pneumonia. Among oral taxa that cause microaspiration [17,18], *Rothia*, *Veillonella*, and *Neisseria* were associated with pneumonia, whereas *Prevotella* and *Streptococcus* were associated with non-pneumonia. Ramnananet al. and Lauferet al. suggested a role for *Rothia* in the pathogenesis of respiratory infection [19,20]. *Prevotella* has been associated with a reduced risk of nosocomial pneumonia [21].

Importantly, the microbial community may be predictive of the clinical outcome in mechanically ventilated critically-ill patients. On day 1, the relative abundances of some taxa, such as *Streptococcus* and *Haemophilus*, were significantly higher in the successful extubation group than in the failed extubation group. We found differences in the composition of the respiratory microbiome according to clinical outcome, such as extubation failure within 3 weeks. The successful extubation group, but not the failed extubation group, showed a change in β-diversity. This association between the longitudinal change in β-diversity and clinical outcome may have been due to host–microbiome interactions. Oral taxa such as *Rothia*, *Prevotella*, and *Streptococcus* were correlated with the change in β-diversity. This finding suggests that some oral taxa play an important role in the clinical outcome of intubated patients with pneumonia.

Network analysis of intubated patients with pneumonia revealed that the abundances of oral taxa (e.g., *Streptococcus*, *Actinomyces*, *Veillonella*, *Granulicatella*, and *Prevotella*) were correlated. Several studies have evaluated oral taxa in the respiratory microbiome [17,22]. Segal et al. reported that aspiration-derived oral microbial species regulate basal mucosal Th17 immune activation in healthy individuals [22]. Piterset al. reported that a model based on the relative abundance of microbes in the upper respiratory tract could differentiate patients with pneumonia from healthy individuals [17].

The association between oral taxa and the outcome of pneumonia is consistent with the concept that oral taxa are associated with a weaker immune response, compared with pathogenic taxa. In a study of mechanically ventilated patients [6], Kitsioiset al. found that 80% of culture-negative respiratory samples had high abundances of oral bacteria, which were associated with less severe lung epithelial injury and inflammation. Therefore, in intubated patients with a suppressed basal respiratory microbiome due to infection, the relative abundances of oral taxa may influence the host response and clinical outcome.

There may be a link between specific taxa in the sputum microbiome and an increased risk of mortality in patients with chronic obstructive pulmonary disease [23,24,25]. The abundance of the oral genus *Veillonella* was significantly decreased during acute exacerbation of chronic obstructive pulmonary disease, as well as in patients who died within 1 year.

The 16S rRNA sequencing data showed that a lower abundance of *Streptococcus* at the time of intubation was associated with 28-day mortality. This result, in conjunction with recent findings, highlights the role of *Streptococcus pneumoniae* in the pulmonary immune response [26,27]; indeed, *S. pneumoniae* colonization reportedly increases the phagocytic capacity of alveolar macrophages [27,28,29]. Mitsiet al. postulated that mechanisms involved with improvement of lung immunity related to *S. pneumoniae* lung persistence may constitute epigenetic changes in alveolar macrophages as a manifestation of adaptive immunity or improved phagocytosis by antigen-stimulated Th1 activation [28]. Clarification of the interactions between bacterial colonization of the respiratory tract and the host immune response could lead to new strategies for prediction and prevention of severe pneumonia requiring mechanical ventilation. In addition, taxa-inferred metagenome data may reveal associations among clinical outcomes, immune status, and microbial metabolism.

There were some limitations in this study. First, it analyzed a small number of patients from a single center. The moderate predictive accuracy of *Streptococcus* (area under the ROC curve, 0.74) for 28-day mortality may be due to the small number of samples, because the area under the ROC curve of the APACHE score (an important risk factor for mortality) was <0.7. Second, this study did not involve a comparison of bronchoalveolar lavage fluid and ETA, because the most recent clinical practice guidelines recommend non-invasive testing via ETA, rather than invasive testing by bronchoscopy [4]. ETA culture is considered the most appropriate and minimal-risk sampling method to investigate the respiratory microbiome of intubated patients in the ICU [7,10]. Third, most patients received at least one dose of antibiotics within 24 h after intubation. Therefore, the effect of antibiotics on the respiratory microbiome was unclear. However, prior studies have reported no association between respiratory microbial diversity and antibiotic treatment [7,9,16].

In summary, we described the composition and diversity of the microbiome of the lower respiratory tract in intubated patients with pneumonia. Importantly, we found a profound longitudinal change in the β-diversity of the respiratory microbiome in successfully extubated patients. This profound longitudinal change was positively associated with the relative abundances of *Rothia*, *Streptococcus*, and *Prevotella*. Metataxonomicanalysis indicated that a low abundance of *Streptococcus* at the time of intubation may be associated with 28-day mortality. Future prospective studies are needed to evaluate the prognostic roles of specific bacterial taxa and their therapeutic potential in the treatment of pneumonia.

## Figures and Tables

**Figure 1 jcm-09-00638-f001:**
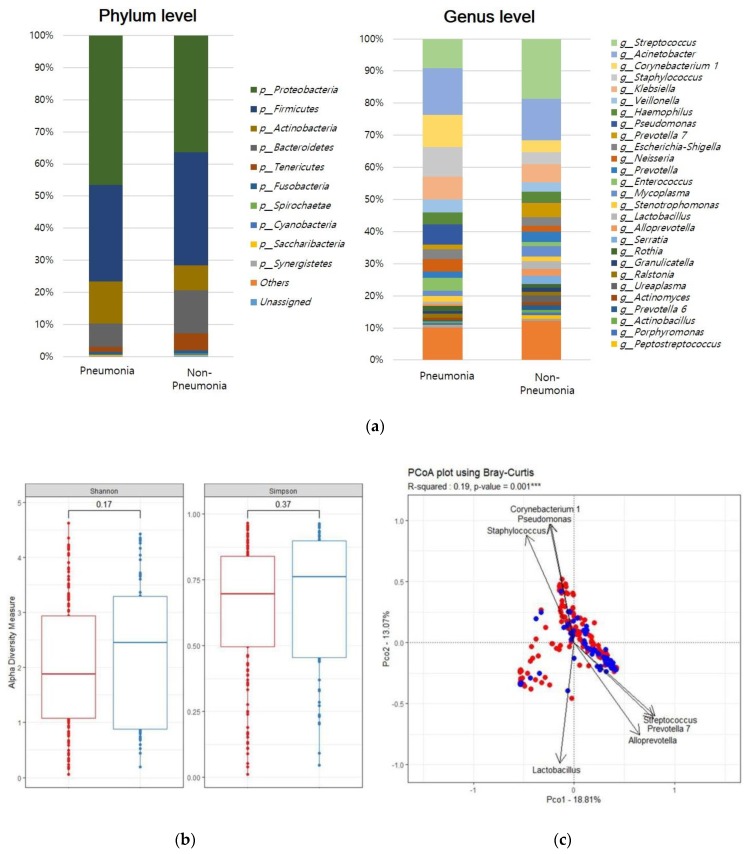
The respiratory microbiome differed between intubated patients with pneumonia and those without pneumonia. (**a**) Lower respiratory tract microbiome composition at phylum and genus levels. (**b**) Shannon diversity index and Simpson index. Red: pneumonia group, blue: non-pneumonia group. (**c**) Bray–Curtis distance of the respiratory microbiota, shown as a principal coordinate analysis (PCoA) two-dimensional map. Each dot represents one sample. (**d**) Histogram of unique biomarker bacteria detected based on the linear discriminant analysis (LDA) effect size (>2.5-fold).

**Figure 2 jcm-09-00638-f002:**
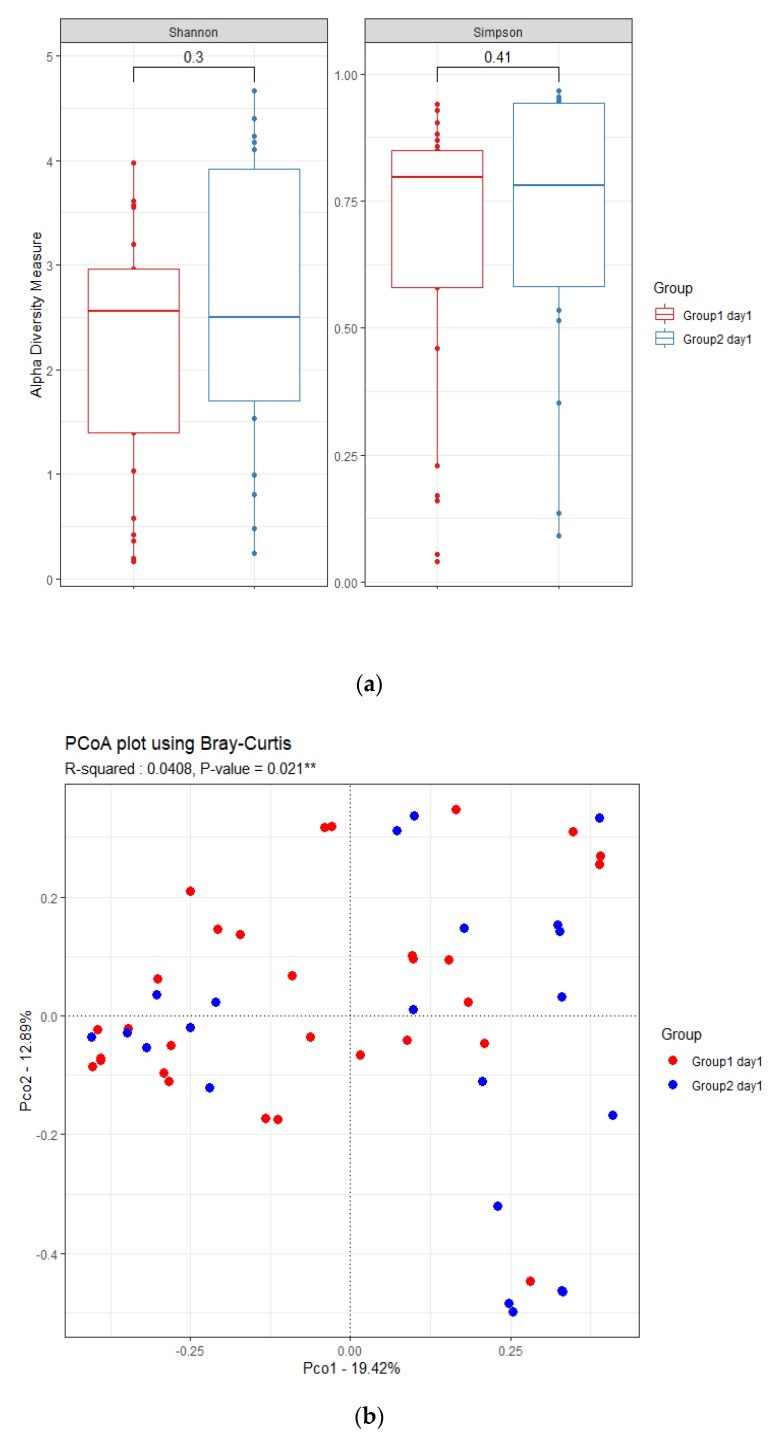
The respiratory microbiome of intubated patients with pneumonia differed between the failed extubation (Group1) and successful extubation groups (Group2). (**a**) Shannon diversity index and Simpson index. (**b**) Bray–Curtis distance of the respiratory microbiota, shown as a PCoA two-dimensional map. Each dot represents one sample. (**c**) Histogram of unique biomarker bacteria detected based on the LDA effect size (>2.5-fold).

**Figure 3 jcm-09-00638-f003:**
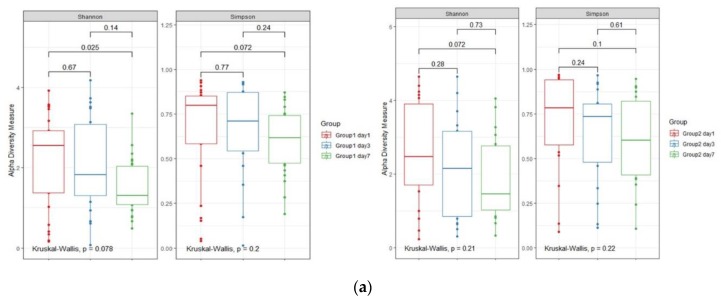
Longitudinal changes in the respiratory microbiota in intubated patients with pneumonia. (**a**) Changes in α-diversity between days 1 and 7 after initiation of mechanical ventilation. The Shannon diversity index and Simpson index are measures of α-diversity. Left, successful extubation group (Group1); right, failed extubation group (Group2). (**b**) PCoA plots of the Bray–Curtis distance of the respiratory microbiota. Each color represents the day. Left, successful extubation group (Group1); right, failed extubation group (Group 2). (**c**) Changes in Bray–Curtis distance are shown as red dashed lines for the successful extubation group, and blue dashed lines for the failed extubation group. Black and gray arrows indicate mean changes in the successful extubation and failed extubation groups between two time points. Logistic regression analysis showed that the changes in PCo2 significantly differed between the two groups (*p* = 0.001).

**Figure 4 jcm-09-00638-f004:**
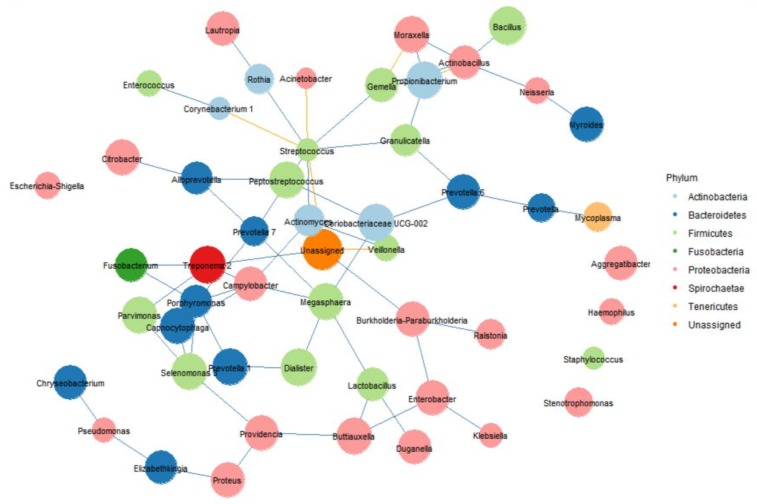
Microbial network of intubated patients with pneumonia, consisting of 53 nodes and 60 edges. Edges show associations with FDR (false-discovery rate) q-values < 0.05 with bidirectionality. Each edge indicates a significant correlation between OTUs; blue, *n* = 54, positive; yellow, *n* = 6, negative.

**Figure 5 jcm-09-00638-f005:**
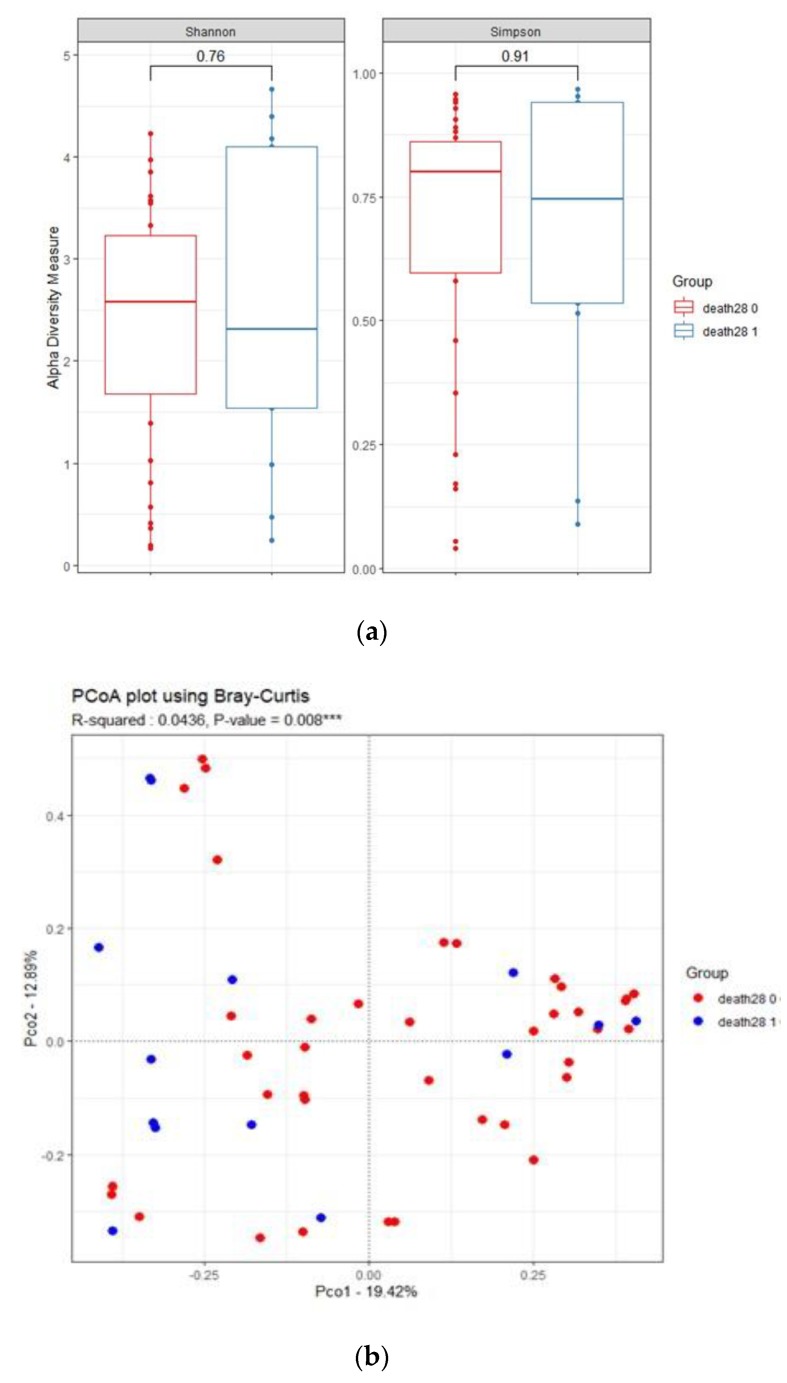
Association between respiratory microbiome and 28-day mortality in intubated patients with pneumonia. (**a**) Shannon diversity index and Simpson index. (**b**) Bray–Curtis distance between samples, shown as a PCoA two-dimensional map. Each dot represents one sample. (**c**) Histogram of unique biomarker bacteria detected based on the LDA effect size (>2.5-fold). (**d**) ROC curve for the association between 28-day survival and the relative abundance of *Streptococcus* in ETA at day 1.LDA, linear discriminating analysis; ROC, receiver operating characteristics; ETA, endotracheal aspirate.

**Table 1 jcm-09-00638-t001:** Demographics and clinical parameters of mechanically ventilated patients in the ICU.

	Pneumonia Group		Non-Pneumonia Group	*p*-Value ^‡^	*p*-Value ^§^
	Successful Extubation	Failed Extubation			
	*n* = 22	*n* = 19	*n* = 19		
Age	72 (58–75)	76 (70–85)	76 (59–81)	0.738	0.063
Male sex ^†^	16 (72.7%)	14 (73.7%)	12 (63.2)	0.547	0.99
Acute respiratory distress syndrome ^†^	2 (9.1%)	6 (31.6%)	0 (0%)	0.047	0.115
Charlson comorbidity index *	2 (0.8–3.3)	3 (2–5)	2(1–2)	0.051	0.227
Cause of intubation ^†^				<0.001	0.269
Cardiac arrest	0 (0.0%)	1 (5.3%)	3 (15.8%)		
Neurological distress	4 (18.2%)	1 (5.3%)	14 (73.7%)		
Postoperative status	0 (0.0%)	0 (0.0%)	1 (5.3%)		
Respiratory	18 (81.8%)	17 (89.5%)	1 (5.3%)		
PaO_2_/FiO_2_	241 (137–325)	188 (102–292)	430 (321, 457)		0.14
Severity *					
APACHE score	17 (15–21.5)	23 (19–25)	22 (17–25)	0.165	0.015
SOFA score	7 (5–8.3)	8 (7–13)	6 (4–9)	0.227	0.045
GCS*	8 (6.8–10)	9 (5–11)	6 (5–9)	0.074	0.895
28-day mortality ^†^	0 (0%)	12 (63.2%)	6 (31.6%)	>0.99	<0.001
ICU mortality ^†^	0 (0%)	17 (89.5%)	6 (31.6%)	0.4	<0.001
MV duration *	8 (7–14)	16 (13–25)	10 (7–14)	0.238	<0.001
CRP (mg/dL) *	139 (65.8–221.3)	133 (44–158)	61 (4.6–132.1)	0.011	0.488

* Median (interquartile range); † frequency (%); ‡ pneumonia vs. non-pneumonia group; § Successful extubation vs. failed extubation group (limited to patients with pneumonia); PaO_2_/FiO_2_: ratio of arterial oxygen partial pressure to fractional inspired oxygen; APACHE II: Acute Physiology, Age, Chronic Health Evaluation II; SOFA: Sequential Organ Failure Assessment; GCS: Glasgow coma scale; ICU: intensive care unit; MV: mechanical ventilation; CRP: C-reactive protein.

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
