# Peer review of "The Dynamics of Respiratory Microbiota during Mechanical Ventilation in Patients with Pneumonia"

_jcm, 2020, doi:10.3390/jcm9030638_

Round 1

Reviewer 1 Report

I was impressed by this paper as it is on an important clinical topic and examines novel mechanisms for survival via the respiratory microbiome. Given that this is a high risk treatment group that are complex to treat these results could inform actual probiotic respiratory treatments once the mechanism and validity of the associations are fully established.

The abstract was clear and covered the key elements of the study

The literature review we well developed and relied on current research and defined the clinical problem well and the importance of examining the respiratory microbiome.

The design and methods section was very detailed and clearly described. There was a great deal of rigor in how the authors developed this study which adds to the defendability of findings.

I thought the scientific analysis method was particularly well described and the testing protocol was appropriate. Likewise, the data analysis was well defined and correct for the various types of data-variables examined.

A detailed and clear results section with good use of graphs for visual illustration of the data

The discussion was well developed and did not go past the findings and integrated other research findings into how the results can be interpreted. Likewise the limitations of the study was realistic and covered the important issues.

Overall, this well written paper was a pleasure to read and is certainly worth of publication.

Author Response

Thank you for reviewing the  paper.

Reviewer 2 Report

The authors present a study characterizing the lung bacteriome for ICU patients during mechanical ventilation.

The analyses were performed using endotracheal aspirate. The authors compared the microbiome of patients at 3 time points (1day, 3days and 7 days post intubation).

The cohort included patients with and without pneumonia and patients were characterized regarding severity (SOFA score, APACHE score, successful extubation after 3 weeks).

Bacterial community characterisation and comparison have been done in order to compare several groups: i/ Pneumonia vs non pneumonia; ii/ Successful extubation vs failed extubation (for pneumonia patients); iii/ Survival/Death after 28 days (for pneumonia patients).

Minor comments:

Editing problem: The Greek letter alpha and beta were not visible in the document (excepted in the abstract) making the reading of the manuscript quite hard.

Lines 20 and 74: Illumina sequencing was performed during this study. It is not pyrosequencing. This should be corrected

Figure 1d: Prevotella appeared 3 times in the graph. Is it a mistake or is it because there is 3 different species and the graph represents genera?

Lines 173, 212 and 249: It seems to me that “Hemophilus” should be spelled “Haemophilus”.

Figure 3: There is incongruence between the text and the figure.

Figure legend: “ (a) Changes in -diversity between days 1 and 7 after initiation of mechanical ventilation. The Shannon diversity index and Simpson index are measures of -diversity. Right, successful extubation group (Group1); left, failed extubation group (Group2). (b) PCoA plots of the Bray–Curtis distance of the respiratory microbiota. Each color represents the day. Right, successful extubation group (Group1); left, failed extubation group (Group 2).”

In the text, line 177 to 181, the alpha and beta diversity are described to be different in the successful extubation group, but on the figure, according to the legend, it seems to correspond to the left panel, so to the failed extubation group. Is there a mistake in the figure legend?

Figure 4: the beginning of the word “Escherichia” is missing at the left of the figure.

Author Response

22th February, 2020

Manuscript Number: jcm-725062

Article Title: The dynamics of respiratory microbiota during mechanical ventilation in patients with pneumonia

Dear editor

We would like to thank all of the editors and reviewers for helping us make a better revision. We revised our manuscript according to the comments and recommendations of the reviewers. We highlighted all changes in the revised manuscript in red letters.

Reviewer reports:

Minor comments:

Editing problem: The Greek letter alpha and beta were not visible in the document (excepted in the abstract) making the reading of the manuscript quite hard.

We revised the manuscript, as suggestions.

Lines 20 and 74:

Illumina sequencing was performed during this study. It is not pyrosequencing. This should be corrected

We revised the manuscript, as suggestions. Pyrosequencing was changed to sequencing.

Figure 1d: Prevotella appeared 3 times in the graph. Is it a mistake or is it because there is 3 different species and the graph represents genera?

This graph represents genera. Prevotella was divided to three subgroups according to SILVA database.

Therefore, to avoid confusions, we combined 3 subgroups to one Prevotella genera.

We revised the Figure 1d.

Lines 173, 212 and 249: It seems to me that “Hemophilus” should be spelled “Haemophilus”.

We revised the manuscript, as suggestions.

Figure 3: There is incongruence between the text and the figure.

Figure legend: “ (a) Changes in -diversity between days 1 and 7 after initiation of mechanical ventilation. The Shannon diversity index and Simpson index are measures of -diversity. Right, successful extubation group (Group1); left, failed extubation group (Group2). (b) PCoA plots of the Bray–Curtis distance of the respiratory microbiota. Each color represents the day. Right, successful extubation group (Group1); left, failed extubation group (Group 2).”

In the text, line 177 to 181, the alpha and beta diversity are described to be different in the successful extubation group, but on the figure, according to the legend, it seems to correspond to the left panel, so to the failed extubation group. Is there a mistake in the figure legend?

Sorry, there was a mistake in the figure legend

We revised the figure legend.

“Figure 3. Longitudinal changes in the respiratory microbiota in intubated patients with pneumonia. (a) Changes in a-diversity between days 1 and 7 after initiation of mechanical ventilation. The Shannon diversity index and Simpson index are measures of α-diversity. Left, successful extubation group (Group1); right, failed extubation group (Group2). (b) PCoA plots of the Bray–Curtis distance of the respiratory microbiota. Each color represents the day. Left, successful extubation group (Group1); right, failed extubation group (Group 2).”

Figure 4: the beginning of the word “Escherichia” is missing at the left of the figure.

We revised the figure4

. Thank you very much for your insightful and thorough advice and provisional acceptance of the manuscript.

Sincerely yours,

---------------------------------------------------------------------------------------------------------------------

Ji Young Hong M.D., Ph.D.

Assistant Professor

Division of Pulmonary and Critical Care Medicine, Department of Medicine, Chuncheon Sacred Heart Hospital, Hallym University Medical Center,

77, Sakju-ro, Chuncheon-si, Gangwon-do 200-704, Republic of Korea.

Tel: 82-33-240-8101 Fax: 033 – 255 – 6244

E-mail: mdhong@hallym.or.kr